Resource competition and coexistence in heterogeneous metacommunities: many-species coexistence is unlikely to be facilitated by spatial variation in resources

Schoolmaster Donald R. Jr schoolm4@msu.edu
W.K. Kellogg Biological Station, Michigan State University , MI , USA
Five Rivers Services at US Geological Survey, National Wetland Research Center , Lafayette, LA , USA
Iwasa Yoh
Electronic publication date: 2013 Aug 22
Publication date: 2013
Volume: 1
Electronic Location ID: e136
Received 2013 Apr 25; Accepted 2013 Jul 30
Copyright: © 2013 Schoolmaster Jr
Copyright year: 2013
Copyright holder: Schoolmaster Jr
License: This is an open access article distributed under the terms of the Creative Commons Attribution License, which permits unrestricted use, distribution, and reproduction in any medium, provided the original author and source are credited.
License URL: https://creativecommons.org/licenses/by/3.0/

Keywords: Metacommunity, Storage effect, Spatial heterogeneity, Resource variation, Competition

Funding: AW Mellon Foundation This work was done while the author was funded under a grant from the AW Mellon Foundation. The funders had no role in study design, data collection and analysis, decision to publish, or preparation of the manuscript.

==============================
There is little debate about the potential of environmental heterogeneity to facilitate species diversity. However, attempts to show the relationship between spatial heterogeneity and diversity empirically have given mixed results. One reason for this may be the failure to consider how species respond to the factor in the environment that varies. Most models of the heterogeneity-diversity relationship assume heterogeneity in non-resource environmental factors. These models show the potential for spatial heterogeneity to promote many-species coexistence via mainly the spatial storage effect. Here, I present a model of species competition under spatial heterogeneity and resource factors. This model allows for the stable coexistence of only two species. Partitioning the model to quantify the contributions of variation-dependent coexistence mechanisms shows contributions from only one mechanism, growth-density covariance. More notably, it shows the lack of potential for any contribution from the spatial storage effect, the only mechanism that can facilitate stable many-species coexistence. This happens because the spatial storage effect measures the contribution of different species to specializing on different parts of the gradient of the heterogeneous factor. Under simple models of resource competition, in which all species grow best at high resource levels, such specialization is impossible. This analysis suggests that, in the absence of additional mechanisms, spatial heterogeneity in a single resource is unlikely to facilitate many-species coexistence and, more generally, that when evaluating the relationship between heterogeneity and diversity, a distinction should be made between resource and non-resource factors.

Introduction

The potential of spatial heterogeneity to promote plant species coexistence is well documented theoretically (reviewed in Amarasekare, 2003), but empirical support documenting the power and scope of its ability to support diverse natural communities is mixed. For example, Lundholm (2009) reviewed 41 observational studies and 11 experimental studies that quantified the relationship between plant species diversity and spatial environmental heterogeneity and found that, while many studies documented positive relationships between the two, the cross-study effect size was not significantly different from zero.

One potential reason for the uncertainty observed in the relationship between plant species diversity and spatial environmental heterogeneity is that the strength of the effect depends on what aspect of the environment is varying; specifically whether it is resource or non-resource factors that vary over space. In experiments and observational studies where a non-resource environmental factor (e.g., soil type, pH) varies, positive relationships between spatial heterogeneity and species diversity are often observed (Reynolds et al., 1997, and many more, reviewed in Lundholm, 2009). However, there is surprisingly little empirical support for strong positive relationships between the degree of spatial heterogeneity in a limiting resource and plant species diversity at the local scale (Stevens & Carson, 2002; Bakker, Blair & Knapp, 2003; Reynolds et al., 2007; Lundholm, 2009). For example, categorizing the factors in the studies reviewed by Lundholm (2009) based upon whether they are resource or non-resources factors, reveals that a significant relationship was found between species diversity and spatial heterogeneity for 71% (49 of 69) of non-resource factors, but only 28.5% (2 of 7) of resource factors.1 These results suggest an important question: should we expect that resource variation should have the same effect on diversity as non-resource variation?

To understand why resource heterogeneity may less effectively facilitate species diversity requires insight into how these factors affect variation dependent species coexistence mechanisms (Chesson, 2000a). If plant species are competing for a common resource, then in a uniform environment, the species that can maintain a positive growth rate at the lowest resource concentration (lowest R∗; Tilman, 1982) is expected to drive all others to extinction. If spatial environmental heterogeneity is to facilitate species coexistence, it must cause variation over space in the identity of the species that has lowest R∗ (Amarasekare, 2003). For the case of non-resource spatial heterogeneity, Chesson (2000a) has identified the three variation dependent coexistence mechanisms that cause variation in competitive ability over space and thus facilitate coexistence: (1) spatial relative non-linearity, which can occur if species have different non-linear responses to a common competitive environment; (2) growth-density covariance, which measures a species’ ability to concentrate its population in the areas that best promote growth (in the absence of competition), and (3) spatial storage effect, which occurs when different species experience best growth in different areas of the environment. Of these three mechanisms, the storage effect is potentially the most important in that it is evoked by many kinds of trait differences among species (making it potentially common) and has been shown to allow the coexistence of many species (Chesson, 1994).

While others have shown that spatial variation in resource supply rates can facilitate coexistence, a similar partitioning of the mechanisms involved has not been reported. One potential reason for the lack of attention to the difference in resource and non-resource spatial variation is that prominent early theoretical papers made assumptions that minimized the differences between them. For example, Tilman & Pacala (1993) and Abrams (1988) published models that assumed the environment consists of discrete patches containing multiple limiting resources, and that there is no, or only limited dispersal between patches, or that dispersal occurs at a time-scale much slower than that of local competitive exclusion. These assumptions approximate a situation where the spatial scale of heterogeneity is much larger than the characteristic dispersal distances of the species in the community (i.e., most dispersal events occur within patches, few between patches). Under these conditions, spatial heterogeneity creates opportunities for species coexistence if each species is the best competitor at some ratio of resource supply rates represented in a subset of patches (i.e., R∗ changes over space). This result is similar to the general result obtained from models of non-resource, environmental heterogeneity based coexistence (Chesson, 2000a).

Recently, however, researchers have begun studying models that assume that competitive exclusion and dispersal occur over similar time-scales (Abrams & Wilson, 2004; Golubski, Gross & Mittelbach, 2008). This assumption approximates the case where the spatial scale of resource heterogeneity is shorter than typical dispersal distances, and is probably more typical of the systems measured in field studies. These models predict that a poor resource competitor may coexist with a better resource competitor, if the better resource competitor experiences more interpatch dispersal (Abrams & Wilson, 2004). Coexistence in this case is possible because dispersal results in a net loss of individuals from the richest patches which in turn reduces the population’s ability to depress resource concentrations in those patches as low as it would in the absence of dispersal; allowing persistence of a competitor that experiences less interpatch dispersal but has a higher R∗ (in a uniform environment). However, the identity and relative strength of the coexistence mechanisms involved are yet to be quantified.

The assumption that competition and dispersal occur simultaneously increases the complexity of the models, leading most researchers to model systems consisting of only a small number of patches (Abrams & Wilson, 2004; Golubski, Gross & Mittelbach, 2008). As a result, these models lack the generality that would allow them to be scaled up to quantify metacommunity-level phenomena such as variation-dependent coexistence mechanisms (i.e., the storage-effect, relative nonlinearity, and growth-density covariance; Chesson, 2000a; Chesson, 2000b). In this article, I derive a simple metacommunity model of plant competition for a single, spatially variable resource. I derive approximate analytical relationships for regional species coexistence from which metacommunity-scale population growth rates may be partitioned into the variation-dependent and variation-independent coexistence mechanisms. These mechanisms are used to argue why spatial variation for resources is less effective than non-resource spatial variation in facilitating coexistence of many species.

Model

The goal of this model is to answer the questions “how many species can coexist via spatial heterogeneity in resource supply and by what means?” To answer these questions, I define a simple model of plants growing in a spatially heterogeneous environment, and then use the framework developed by Chesson (1994) to partition the regional growth rates implied by the model into contributions from variation-independent and variation-dependent mechanisms. Finally, I use a sequential invasion approach with the mutual invasion criterion to determine how many species each mechanism allows to coexist at equilibrium.

In a set of discrete patches, x = {1, 2, 3, …, N}, let njx(t) be the density of species j in patch x at time t, and let Rx(t) be the resource concentration in patch x. njx decreases over time at the per capita mortality rate mj, and increases by reproduction in the current patch, which occurs at the resource-dependent per-capita growth rate bjRx, plus the contributions of dispersal into patch x from other patches. Let pj be the proportion of seeds produced by adults of species j in any patch that remain in that patch and assume that the 1−pj seeds that leave natal patches are evenly redistributed among all patches (including the natal patch). Resources are increased in patches at a constant rate Sx and are reduced through the establishment of plants. The resource model is kept intentionally simple to allow analytical treatment. The inclusion of additional loss terms for resources, for example to adult plant maintenance or leaching, do not affect the conclusions (Supplemental Appendix S2). The dynamics of this coupled system are described by, (1a-b) dnjxdt=−mjnjx+bjRx(pjnjx+(1−pj)〈nj〉x)dRxdt=Sx−∑jQjbjRx(pjnjx+(1−pj)〈nj〉x).

In Eqs. (1), Qj is the amount of resource required for establishment; Sx is the patch-specific resource supply rate and 〈⋅〉x indicates a mean taken over patches. For a single species, this system has one stable equilibrium point per patch at, (2a-b) njx*=SxQjmj,Rjx*m=mjSxbj(pjSx+(1−pj)〈S〉x),

where the m in Rjx*m is used to differentiate this equilibrium concentration of resource from the traditional R∗, that occurs in an uncoupled or homogenous system and is independent of resource supply rate.

Equation (2) shows that, in this model, dispersal has no effect on equilibrium density, njx∗. However, the equilibrium resource concentration, Rjx*m does depend on the amount of dispersal between patches and the resource supply rate in patch x relative to the mean supply rate in the metacommunity. Specifically, if dispersal between patches is high, species j leaves a higher concentration of resources behind in patches with above average supply rates and a lower concentration in patches with below average supply rates than it would in a homogenous environment. This occurs because patches with high supply rates are net exporters of recruits and patches with low supply rates are net importers of potential recruits. The increased concentration of available resources in high supply rate patches allows the invasion and possible coexistence of a species that has a higher R∗, but experiences less interpatch dispersal (Abrams & Wilson, 2004) (Fig. 1).

Figure 1 R∗ as a function of dispersal.

Concentration of resources left behind at equilibrium (Rjx∗m) along a spatial gradient of resource supply rates. The Rjx∗m of a species depends on the supply rate of the patch and the amount of interpatch dispersal (Black lines: R∗ = 0.4, p = 1, solid; p = 0.66, dashed; p = 0.33, dash-dot; p = 0, dotted). As a result, a species with a higher R∗ may invade the metacommunity if it experiences less interpatch dispersal (Grey line: R∗ = 0.5; p = 0.9) because it can have a Rjx∗m lower [i] in patches with the highest supply rates.

Derivation of variation dependent mechanisms

One way to measure potential for coexistence is with the mutual invasion criterion. This criterion states that a set of species can coexist with one another if each can invade the equilibrium assemblage of the other species in the set. In practice, one calculates growth rate of a species with interspecific competition set at the value determined by the competitors at equilibrium and intraspecific competition set to zero. This is called the low-density growth rate of the species. In a spatial context, we are interested in coexistence at the larger scale of the set of all patches, so we calculate what is called the low-density metacommunity scale (or regional) growth rate.

Chesson (2000a) has shown how a low density metacommunity-scale growth rate can be calculated and written in terms of variation dependent coexistence mechanisms. First, the local growth rate is decomposed into terms quantifying the direct effects of environmental variation (E), variation in competition (C) and their interaction (3) rjx=Ex−Cjx+γjExCjx,

where Ex=Gj(Ex,Cj*), Cjx=−G(E*,C), γj=∂Gj∂E∂Cj.

The quantities E and C are the population parameters affected by environmental variation and the effect of competition (which is also affected by environmental variation) respectively, G is the growth rate of species j as a function of E and C and the (∗) indicates the equilibrium level of the value.

The metacommunity-scale growth rate of species j, rj˜, in a spatially heterogeneous environment is found by taking the mean of rjx over all individuals in the metapopulation, rj˜=1∑xnjx∑xrjxnjx. It can be written in terms of a spatial mean by defining relative local density as νjx=njx〈nj〉x. Substituting νjx into rj˜ gives rj˜=〈rjνj〉x, which can be rewritten as rj˜=〈rj〉x+cov(rj,νj)x, where cov(⋅)x indicates a spatial covariance (Chesson, 2000a). Plugging Eq. (3) into this result gives, (4) rj˜=〈E〉x−〈Cj〉x+γj〈ECj〉x+cov(rj,νj)x.

To argue that any of the terms in Eq. (4) contribute to coexistence, the difference between invader and resident values must be positive. Because, by definition, the metacommunity growth rate of the resident is zero, it can be subtracted from the right side of Eq. (4) without changing the left side. Thus, the metacommunity growth rate of the invader, denoted by subscript i, can be rewritten in terms of contributions from multiple coexistence mechanisms by subtracting the metacommunity growth rate of the residents, denoted by subscript r. Subtracting the metacommunity growth rate of the residents gives, (5) ri˜=ΔE−ΔC+ΔI+Δκ

where (6) ΔE=〈Ei〉x−qir〈Er〉xΔC=〈Ci〉x−qir〈Cr〉xΔI=γi〈ErCi〉x−qirγr〈ErCr〉xΔκ=cov(ri,νi)x−qircov(rr,νr)x

and the scaling factor qir=∂Ci∂Cr is chosen to make the resulting expression more biologically interpretable. For example, in this case, it allows ΔC to be expressed as a difference in the R* values of the species.

The quantity ΔE measures differences in the average environment experienced by the invader and resident. The quantity ΔC can contain both the fluctuation independent difference of the average competition experienced by residents and invaders and a measure of the effect of variation in competition. The combination ΔE−ΔC is often rewritten to separate it into variation dependent ΔN and variation independent ri˜′ parts, where ri˜′=ΔE−Ci* and ΔN=ΔC−Ci*. The quantity Ci* is the value of competition the invader experiences as a consequence of the resource equilibrium the residents create (Chesson, 2000a). The quantity ri˜′ is the growth rate the invader would experience in the absence of variation. The quantity ΔN has been called relative-nonlinearity and can facilitate coexistence if species exhibit different non-linear responses in growth rate to variations in competition, specifically if the species with the larger non-linearity in response experiences lower variance in competition. For most models that would describe the growth of plants, this mechanism can only facilitate coexistence of two species (Chesson, 1994).

The quantity ΔI is the storage effect (Chesson, 1994). It measures the covariance between the direct effect of environmental variation and the effect of competition on the growth rate of the invader. This mechanism is potentially very powerful and can facilitate coexistence on many species. An example of many species coexistence via spatial storage effect is where there are many patches and each species is the best competitor in at least one patch (Sears & Chesson, 2007).

The final mechanism, Δκ, is growth-density covariance. It measures the ability of the invader to concentrate its population into patches that are best at supporting growth. The species that is better able to do this will experience an overall boost to metacommunity scale growth rate. This mechanism is most directly related to dispersal.

Derivation of variation dependent mechanisms for spatial resource heterogeneity

Following Chesson (2000a) and Chesson (2000b), the model (Eqs. (1)) can be written in terms of variation in environment, Ex (the life history character that varies in space) and competition, Cx. In the case of the present model, where only the supply rate of resources varies over space, environmental variation does not directly affect individuals, it only affects competition; thus Ex = 0 and Cx = −Rx. Making these substitutions, the model in Eqs. (1) can be rewritten as, (7) dnjxdt=−mjnjx−bjCx(pjnjx+(1−pj)〈nj〉x)dCxdt=−Sx+∑jQjbjCx(pjnjx+(1−pj)〈nj〉x).

The local growth rate is then, rjx = −mj−bjpjCx−(1−pj)bj〈C〉x. Notice that in this case, due to dispersal between patches rjx≠1njxdnjxdt. Instead, rjx is derived by considering the fitness of an individual in patch x (Miller & Chesson, 2009). Taking the mean over space, as described in the previous section gives the metacommunity-scale growth rate, ri˜=−〈Ci〉x+cov(ri,νi)x, where 〈Ci〉x=mi+bi〈C〉x. Assuming a single resident, denoted by subscript r, allows calculation of the various coexistence mechanisms for the case of two species. This is helpful to determining the presence or absence of the various mechanisms implied by this model. Plugging into Eq. (6) and scaling by bi to put the results in the natural time scale of generations instead of an absolute time scale (e.g., months or years) (Chesson, 2008) gives (for details see Supplemental Appendix S1), (8) ΔEbi=0ΔCbi=−mibi+mrbrΔIbi=0Δκbi≈mrbr(1−pr)2〈s2〉xpi1−pi−pr1−pr.

Rewriting ΔE−ΔC terms of ri˜′ and ΔN gives, (9) ri˜′bi=−Ri*+Rr*,Δκbi≈var(s)x(1−pr)2Rr*pi1−pi−pr1−pr

where mjbj=Rj*. Notice in this case ΔN = 0 since ΔC contains no variation dependent terms. Biologically, ri˜′/bi is a comparison of abilities to reduce resources in a homogenous environment. Specifically, Eq. (9) states that invaders are benefited by the ability to reduce resources to a lower level than the resident in a homogenous environment (Tilman, 1982). It also suggests that in the absence of spatial heterogeneity (var(s)x = 0) coexistence is impossible because only the species with the lowest R∗ would have a positive growth rate as an invader.

In addition, Eq. (9) shows that only differences in dispersal can allow coexistence in this model. The storage effect (ΔI) and relative nonlinearity (ΔN) have no effect on an invader’s metacommunity-scale growth rate and thus do not contribute to coexistence. The effect of growth-density covariance (Δκ) on the growth rate of the invader depends upon Rr∗ times the odds of seeds staying in natal patches relative to the resident’s. Invaders whose seeds are less widely dispersed than those of the resident species are benefited by growth-density covariance. Thus, this model allows coexistence of species if there is a tradeoff competitive ability (R∗) and dispersal fraction (p). In addition, coexistence is facilitated by only one variation-dependent mechanism, growth-density covariance.

How many species does growth-density covariance support?

I used a sequential invasion approach based on adaptive dynamics (Geritz et al., 1998) to determine how many species can coexist based on this model. This approach attempts to find the trait values for invaders that allow invasion in a given context. In this case, at each step, I find the values for the trait pi that allows an invader to have a positive metacommunity growth rate (often written as S(pi) in the adaptive dynamics literature) in an assemblage of n resident species, i.e., S{prn}(pi) > 0. If this species can (1) coexist with the current resident strategies and resist exclusion by similar strategies (i.e., similar values p), then it is added to the list of residents. Species are added one at a time until there is no value of pi that leads to positive metacommunity-scale growth rate. Because the approximated expression for growth-density covariance from Eq. (9) assumes small variation in supply rate and has singularities at pi = 1 and pr = 1, I use the simulation of Eqs. (1) to calculate invasion growth rate. Numerical simulations of the residents were run until they reached a steady state. The resulting value of resource concentration was plugged into the matrix that described each patch’s contribution to an invader’s low-density growth-rate in patch x. The dominant eigenvalue of this matrix is an estimate of the metacommunity scale low-density growth rate of the invader. For the simulations, I assumed Log-normal distributed supply rates and trade-offs between Rj∗ and pj of the form Rj*−pjτ=Z where Z is an arbitrary constant and τ affects the shape of the trade-off. The following example considers a linear tradeoff, between competitive ability and dispersal i.e., τ = 1.

Figure 2 shows a contour plot for the invasibility of strategy pi in the presence of resident pr. The black region of Fig. 2 represent negative invader growth rates (i.e., the invasion is unsuccessful), the white regions show areas of positive growth rates. It shows that any resident strategy pr < 1 can be invaded by pi = 1. However, the strategy pr = 1 is also able to be invaded by any pi < 1. Thus, I set pr1 = 1 and looked for values of pr2 that could coexist with pr1 and were not excluded by other similar strategies. Figure 3 shows that given pr1 = 1, any resident strategy pr2 > 0 can be invaded by pi < pr2. Thus, since pr2 = 0 can also coexist with pr1 = 1, it was added to the resident list. Figure 3 also suggests that there is no third strategy pi that can coexist with pr1 = 1 and pr2 = 0, since S{pr1=1,pr2=0}(pi) < 0 for all possible values of pi (i.e., a vertical line drawn from pr2 = 0 passes only through the black region of the graph). This suggests that, in this model, a tradeoff between competitive ability Rj∗ and dispersal pj, allows coexistence of only two species. To assure these results were robust, I simulated systems of simultaneous competition among many species from along the trade-off manifold (Supplemental Appendix S2). Those simulations also show an observed maximum of two coexisting species for nonlinear tradeoffs with τ < 1; additional loss terms in the resource equation, and saturating growth responses. For non-linear tradeoffs with τ > 1 no coexistence was possible.

Figure 2 Invasion analysis.

Contour plot of the growth rate of an invader with dispersal pi as a function of resident dispersal pr, assuming a tradeoff between competitive ability (R∗) and dispersal (p). Black regions show areas of negative invader growth rate; white regions, positive. The graph shows that pi = 1 can invade a metacommunity with a resident with pr < 1. It also shows that pr = 1 can be invaded by any pi < 1, suggesting coexistence with this strategy is possible. Parameter values for this graph are Rj∗=1−pj in a system of 20 patches and Sx ∼ LogNormal (1, 1.5).

Figure 3 2 species invasion analysis.

Contour plot of the growth rate of an invader with dispersal pi as a function of resident dispersal pr1 = 1, pr2, assuming a tradeoff between competitive ability (R∗) and dispersal (p). Black regions show areas of negative invader growth rate; white regions, positive. The graph shows that given pr1 = 1, pr2 > 0 can be invaded by and excluded by pi < pr. Thus, it shows that given pr1 = 1 and pr2 = 0, no third strategy pi has a positive growth rate; confirming that stable coexistence of only two species is possible. Parameter values for this graph are Rj∗=1−pj in a system of 20 patches and Sx ∼ LogNormal (1, 1.5).

Discussion

This analysis shows that spatial variation in resources allows for fewer coexistence mechanisms and lower potential species diversity compared to spatial variation in non-resource factors. Although spatial resource variation promotes coexistence if there is a tradeoff between competitive ability and the ability to retain offspring in good patches, this tradeoff allows the coexistence of only 2 species.

Coexistence under spatial resource variation lacks contributions from the spatial storage effect, a powerful mechanism that allows the coexistence of many species. The storage effect is absent because all species grow best in the same patches (high resource patches). As a result, environmental responses and competition are perfectly and equally correlated for all species, allowing no advantages in good patches when a species is rare. In other words, the feedback that organisms have on the resource make it impossible for any to specialize on a particular supply rate along a gradient.

Although the mechanism of resource competition analyzed in this model is simplified, the qualitative results are quite general. For example, the relationship of R∗ to the metacommunity-scale coexistence mechanisms described by scale transition theory does not depend on the simplified model of local resource competition presented in Eqs. (1). If the establishment rate of seeds is a saturating function (e.g., Monod functions with equal half saturation constants) of the amount of resources present, then r′˜ is still a comparison of the (more complicated) R∗s of the invader and resident, and the relative importance of the variation dependent mechanism is the same. The situation is more complex if species growth rates are different non-linear functions of resource concentration but, even in this case, spatial variation in one only facilitates 2 species coexistence (Supplemental Appendix S2).

In this model, competition was for a single resource. Increasing the number of limiting resources can increase the number of species that can coexist at equilibrium. Golubski, Gross & Mittelbach (2008) found that a maximum of four species could coexist in a system with two resources, heterogeneous supply rates and species that were capable of integrating growth across patches. However, this was only possible with precise arrangement of species trait parameters, and if resource supply rates of the resources were strongly negatively correlated across patches. I presume that, in this case, adding resource factors does not increase the number of mechanisms; the existing mechanism, growth-density covariance, simply works independently for each resource; although this claim should be explored with further analysis.

This analysis suggests that spatial resource heterogeneity is not capable of supporting the robust stable coexistence of many plant species that is observed in many natural systems. The reason for this is that the feedback that species have on resource concentrations prevents specialization of different species at different supply rates. It is this kind of specialization of different species along different points of a “niche” axis, which is measured by the spatial storage effect, that allows the robust coexistence of many species. The results of this analysis are consistent with the patterns found in the empirical literature, which finds much more support for the relationship between species diversity and nonresource spatial heterogeneity than resource heterogeneity. Taken together, this work suggests that the consideration whether a factor is a resource or not is crucial for those attempting to understand real patterns in species diversity or those interested in managing a habitat for increased species diversity.

Supplemental Information

Supplemental Appendix S1 Supplemental Appendix S1

Derivation of standardized density vjx.

Click here for additional data file.

Supplemental Appendix S2 Supplemental Appendix S2

Simulation of Many-Species Dynamics.

Click here for additional data file.

I would like to thank Peter Chesson, Kay Gross, and Gary Mittelbach for helping improve this manuscript. This is contribution No. 1725 from the W.K. Kellogg Biological Station, Michigan State University.

Additional Information and Declarations

Competing Interests

Author Contributions

1 Factors related to water (e.g., soil moisture) were left out of these tallies since they can often act as both resource and non-resource factors.

Donald R. Schoolmaster Jr is an employee of Five Rivers Services.

Donald R. Schoolmaster Jr conceived and designed the experiments, performed the experiments, analyzed the data, contributed reagents/materials/analysis tools, wrote the paper.

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
