# Peer review of "Resource competition and coexistence in heterogeneous metacommunities: many-species coexistence is unlikely to be facilitated by spatial variation in resources"

_PeerJ, doi:10.7717/peerj.136_

## Round 0.1 · original submission · Major Revisions

Please read the comments and resubmit a fully revised version which will be sent back to the same two reviewers.

Reviewer 1 ·

Basic reporting

No comments

Experimental design

The manuscript begins by pointing out that empirical studies may overlook the distinction between spatial heterogeneity in non-resource factors from heterogeneity in resource factors as potential drivers of competitive coexistence. This makes a lot of sense, though I wonder if resource factors other than water are not also likely to act as (or at least correlate closely with) another non-resource factor. I felt that the introduction's review of the relevant theory could have been strengthened: the Abrams and Wilson and Golubski et al. papers appear to constitute the only reasons one might think that variation in a single limiting resource could facilitate coexistence, but they are not described in enough detail for the reader to understand the mechanism by which it could occur. Moreover, no multiplicity of situations where such heterogeneity facilitates coexistence appears to exist, so it's not clear that Chesson-like classification of mechanisms is needed.

By the end of the paper, it seems that the study is really more about growth-density covariance, and how many species this mechanism can support, than about resource vs. non-resource heterogeneity. I felt that this frame might relieve some of the concerns initially I had (above) over why one would be interested in resource heterogeneity. In any case, I really would have appreciated a clear, intuitive description earlier on (perhaps at the end of the model section) of how the processes described constitute growth-density covariance (although this will be shown later on in the paper). If the main thrust of the paper is that growth-density covariance can maintain at most two species, I think that result should be highlighted.

I think the model sections lack some justification and detail. First, the constant supply rate is odd, as in the absence of consumers, the resource grows linearly. Either exponential growth or a proportional loss term would strike me as more reasonable. Second, the rest of the model (for a single patch) appears to be a reparameterization of a Lotka-Volterra consumer-resource model. I felt it would be more straightforward to describe it as such; the current text, which uses terms of reproduction and establishment, make it sound as if resources are not use to maintain biomass, but only to produce new biomass. I could have used some steps on the way to Equations 2; E* and C* in Eqns 3 do not seem to be defined; g_j and F_j in Equations 8 are not defined and the text below refers to b_j, which does not appear in equations 8; is it correct that F_j does not appear in these same growth rates below Eqn 8?; it would be helpful to describe m_i/b_i as R* after the second set of equations labeled (8) on page 10, as it is not done otherwise; and the equation numbering should be fixed as there is a reference to equation (10).

Minor points: on page 3, relative nonlinearity is described as involving different responses to a common competitive environment, which is a little confusing as Chesson's scheme is presented as addressing non-resource variation only. Also, it is difficult from the following descriptions to distinguish growth-density covariance from the spatial storage effect.

Validity of the findings

The model section and Figure 1 certainly helped me to understand how resource heterogeneity might enable coexistence. The figure and discussion, however, do seem to imply that many species might coexist on a continuum of dispersal-R* tradeoff (analogous to the competition-colonization tradeoff). But the simulations maximize growth rate, so that the first invader p_1=0 and the second p_2=1. This leaves no room for a third species with higher R* in rich environments but more restricted dispersal. Since any second species with p_2>0 can invade, did the author try a case with 0<p_2<1 and p_3<p<2? If not, I also wonder if exponential resource growth might facilitate coexistence by increasing absolute supply in rich patches (although exponential resource growth seems more realistic for animals than for plants.)

Reviewer 2 ·

Basic reporting

no comment

Experimental design

no comment

Validity of the findings

Extracted from the general comments:

p. 11: "Species are added one at a time until there is no value of p_i
that leads to [a] positive growth rate." Does the order of
introduction matter? Does it matter what p you start with? I'm
thinking that the answer to these questions is no but only because
I've read the supplemental material and seen that Fig. B1 shows
the dynamics of 50 species introduced simultaneously. I found that
test more convincing, though it's hardly conclusive.

p. 12: Why were invaders chosen to maximize the invasion growth rate?
There's no optimization process going on here: either an invader
succeeds or it doesn't, and if it succeeds, it becomes a new resident,
either supplanting the old one or coexisting with it. There aren't
repeated trials of invaders, with the best one remaining in the system
(while the original resident is unaffected by the process). I assume
this was done to prove some sort of generality but the logic escapes
me. Again, without seeing the supplementary material I was highly
skeptical of the conclusions of this paper.

p. 12: "In a system with two residents at p_{r1} = 1 and p_{r2} = 0,
there is no dispersal strategy that can invade..." When I initially
read this, I thought that this conclusion was supposed to follow from
the previous statements, which is nonsense. Reading the supplement, I
see that more work was done. It's critical to at least point the
reader at the relevant portion of the supplement, and ideally, that
work would be discussed in the main body.

p. 13: "The storage effect is absent because all species grow best in
the same patches (high resource patches)." What if there is more than
one resource, a la Tilman? Can we get a storage effect then?

Additional comments

I strongly encourage the author to spend more time proofreading. The
number and severity of typos in this ms was frustrating.

******* Most worrisome typos *********

p. 9: "...if the growth rate if the species exhibit..." After a few
readings, I decided that this probably was supposed to read something
like "... if the species growth rate exhibits...."

p. 10, eq. 8: where do g and F come from? I think g is supposed to be
b. I don't know what F is. This equation also doesn't agree with the
subsequent equation in the text.

p. 11: all references to eq. 10 appear to actually be references to
eq. 9.

***** Greater concerns *********

p. 8, paragraph after eq. 4: I found this paragraph really confusing.
I recommend first pointing out that the resident's growth rate is
zero, then noting that this means that if any of the terms in eq. 5 is
to contribute to coexistence, the difference between the invader's and
residents' values must be positive. And why scale both to be in units
of resources? I know Chesson does it (to get rid of the linear term
in delta C), but the benefit of the rescaling doesn't follow from
anything you've said in this paragraph.

p. 10: "Plugging into eq. 6 and scaling by b_i..." Why scale by b_i?
Explain in a few words and then give the reference rather than asking
the reader to read Chesson 2005.

p. 11, eq. 9: The r' comes out of nowhere. The way the ms reads now,
it sounds as though the reader is supposed to recognize r' and not
take it as a new definition --- and I do recognize it as
Chesson's usual r', but not many know Chesson's work that well. I
think you can avoid getting deep into Chesson lore and say something
like, making the definition r'/b = blah, we can re-write \tilde{r}/b
as the sum of blah and blah. Or do these sum to \tilde{r}? Honestly,
it's not clear where these come from.

p. 11: "Species are added one at a time until there is no value of p_i
that leads to [a] positive growth rate." Does the order of
introduction matter? Does it matter what p you start with? I'm
thinking that the answer to these questions is no but only because
I've read the supplemental material and seen that Fig. B1 shows
the dynamics of 50 species introduced simultaneously. I found that
test more convincing, though it's hardly conclusive.

p. 12: Why were invaders chosen to maximize the invasion growth rate?
There's no optimization process going on here: either an invader
succeeds or it doesn't, and if it succeeds, it becomes a new resident,
either supplanting the old one or coexisting with it. There aren't
repeated trials of invaders, with the best one remaining in the system
(while the original resident is unaffected by the process). I assume
this was done to prove some sort of generality but the logic escapes
me. Again, without seeing the supplementary material I was highly
skeptical of the conclusions of this paper.

p. 12: "In a system with two residents at p_{r1} = 1 and p_{r2} = 0,
there is no dispersal strategy that can invade..." When I initially
read this, I thought that this conclusion was supposed to follow from
the previous statements, which is nonsense. Reading the supplement, I
see that more work was done. It's critical to at least point the
reader at the relevant portion of the supplement, and ideally, that
work would be discussed in the main body.

p. 12: There's a tradeoff of the form m_j / b_j - p_j^tau = C. First,
please don't re-use C. That's confusing. Second, I found the choice
of tradeoff quite puzzling until I read the appendix and realized that
the intent was to create a trade-off between competitive ability and
dispersal ability. I couldn't understand why we should associate low
mortality with high dispersal or high birthrate with low dispersal.
(But perhaps it's reasonable if high dispersal probability requires
the creation of specialized dispersal structures, which takes away
from the resources needed for survival or reproduction?) The
confusing point is that death and birth rates aren't just about
competitive ability. I encourage you to make the intent of the
tradeoff more clear.

p. 13: "The storage effect is absent because all species grow best in
the same patches (high resource patches)." What if there is more than
one resource, a la Tilman? Can we get a storage effect then?

*** Lesser concerns **********

p. 4 Why does having competitive exclusion occur on the same timescale
as dispersal approximate the case where the spatial scale of resource
heterogeneity is shorter than typical dispersal distances? That's not
obvious to me.

---

## Round 0.2 · accepted · Accept

The revised version is suitable for the publication in PeerJ. Thank you for submitting your work in Peer J.

Reviewer 1 ·

Basic reporting

No comments

Experimental design

I think the author should make clear in the text that Supplemental Appendix 2 addresses the effect of a proportional resource loss in the model using a simulation approach.

Validity of the findings

The author has adequately addressed my previous concerns regarding the validity of the invasion analysis.

Additional comments

This article now meets the PeerJ criteria for publication in my opinion.